# Culturable and Non-Culturable Blood Microbiota of Healthy Individuals

**DOI:** 10.3390/microorganisms9071464

**Published:** 2021-07-08

**Authors:** Stefan Panaiotov, Yordan Hodzhev, Borislava Tsafarova, Vladimir Tolchkov, Reni Kalfin

**Affiliations:** 1National Center of Infectious and Parasitic Diseases, 1504 Sofia, Bulgaria; y.hodzhev@ncipd.org (Y.H.); b.tsafarova@ncipd.org (B.T.); tolchkov@ncipd.org (V.T.); 2Institute of Neurobiology, Bulgarian Academy of Sciences, 1113 Sofia, Bulgaria

**Keywords:** blood microbiome, metagenomics, targeted sequencing

## Abstract

Next-generation sequencing (NGS) and metagenomics revolutionized our capacity for analysis and identification of the microbial communities in complex samples. The existence of a blood microbiome in healthy individuals has been confirmed by sequencing, but some researchers suspect that this is a cell-free circulating DNA in blood, while others have had isolated a limited number of bacterial and fungal species by culture. It is not clear what part of the blood microbiota could be resuscitated and cultured. Here, we quantitatively measured the culturable part of blood microbiota of healthy individuals by testing a medium supplemented with a high concentration of vitamin K (1 mg/mL) and culturing at 43 °C for 24 h. We applied targeted sequencing of 16S rDNA and internal transcribed spacer (ITS) markers on cultured and non-cultured blood samples from 28 healthy individuals. Dominant bacterial phyla among non-cultured samples were Proteobacteria 92.97%, Firmicutes 2.18%, Actinobacteria 1.74% and Planctomycetes 1.55%, while among cultured samples Proteobacteria were 47.83%, Firmicutes 25.85%, Actinobacteria 16.42%, Bacteroidetes 3.48%, Cyanobacteria 2.74%, and Fusobacteria 1.53%. Fungi phyla Basidiomycota, Ascomycota, and unidentified fungi were 65.08%, 17.72%, and 17.2% respectively among non-cultured samples, while among cultured samples they were 58.08%, 21.72%, and 20.2% respectively. In cultured and non-cultured samples we identified 241 OTUs belonging to 40 bacterial orders comprising 66 families and 105 genera. Fungal biodiversity accounted for 272 OTUs distributed in 61 orders, 105 families, and 133 genera. Bacterial orders that remained non-cultured, compared to blood microbiota isolated from fresh blood collection, were Sphingomonadales, Rhizobiales, and Rhodospirillales. Species of orders Bacillales, Lactobacillales, and Corynebacteriales showed the best cultivability. Fungi orders Tremellales, Polyporales, and Filobasidiales were mostly unculturable. Species of fungi orders Pleosporales, Saccharomycetales, and Helotiales were among the culturable ones. In this study, we quantified the capacity of a specific medium applied for culturing of blood microbiota in healthy individuals. Other culturing conditions and media should be tested for optimization and better characterization of blood microbiota in healthy and diseased individuals.

## 1. Introduction

In the course of evolution, many microbial species have successfully adapted to the human macroorganism. According to the results of the “Human Microbiome Project”, 90% of the cell mass of the body in adults is composed of microorganisms [1]. Quantitatively, this is about 40 trillion cells [2]. A recent catalog based on microbial taxonomic units from 16S rRNA genes and metagenomically assembled microbial genomes identified at least 5000 species inhabiting the human body [3,4], or up to 10,000 according to others [5]. Most microorganisms are critical to normal human health. A significant part of the human normal microbial flora is not culturable and was indirectly identified by DNA sequencing methods. Blood and internal organs and tissues of healthy people are considered a “sterile” environment due to the lack of proliferative microorganisms. A recent study demonstrated in more than 1500 tumor samples, including adjacent normal tissues, a rich microbiome composed of 528 bacterial species mostly intracellular [6]. As a rule, all conditions in which bacterial flora is isolated from the blood are considered pathological.

During the last half-century, the scientific community started a dynamic debate concerning the existence of normal microbial flora in the blood of healthy individuals [7]. The accepted dogma that blood is a sterile environment is under revision and divided researchers into three groups. The first one supports the hypothesis, that blood is not as sterile as previously supposed and that blood microbiota are naturally existing in the blood of healthy individuals [8,9,10,11,12,13,14,15]. Dormant, latent, or non-culturable microbial forms should be considered [16,17]. The second assumes that identified microbial DNAs in the blood are cell-free circulating DNAs [18,19,20]. The third one denies the existence of enigmatic blood microbiota in healthy individuals [21,22].

One obvious question needs to be asked: why the surfaces of skin, eyes, ears, nose, mouth, and gastrointestinal tract which are inhabited by billions of microorganisms are healthy, while the internal organs and blood are considered sterile, and any contact with microbial species should be considered as deleterious, leading to disease?

Until recently it was only accepted that blood is sterile and that existence of circulating blood microbiota in healthy individuals is associated with transitory microbial circulation due to surgery, dentist manipulations, or state of infection after skin injury. At the same time, it is proven that in the blood of clinically healthy individuals, microorganisms could persist for many years without causing illness, like latent tuberculosis and many other latent infections. Evidence for the existence of healthy blood microbiota is steadily accumulating. Viruses, such as torque teno virus (TTV), anelloviruses, astroviruses, pegiviruses (human PgV), and others, are commonly detected in blood donors [23,24,25] without inducing any clinical symptoms. This state is defined as asymptomatic acute viremia. Growth of bacterial flora in blood erythrocyte suspension was suspected by Guido Tedeschi in 1969 and confirmed by radiometric methods as the incorporation of nucleosides in human erythrocytes, attributed to the metabolic activity of mycoplasma or bacterial L-forms [26]. In 1977, Domingue and Schlegel identified in 7% of the blood specimens from supposedly healthy individuals novel bacterial structures [8]. In 1993 our colleague Emil Kalfin experimentally proved by culturing and electron microscopy that microorganisms are multiplying in the erythrocytes of healthy individuals [10]. Kalfin reported 100% positivity of the blood cultures. Kalfin’s research was continued by us, and in 2018 we demonstrated by culture, targeted sequencing, light and electron microscopy the existence of rich bacterial and fungal diversity in the blood of healthy individuals [12].

This study fills in a knowledge gap and provides an analysis of the cultivability of blood circulating microbiota.

## 2. Materials and Methods

### 2.1. Culturing

Blood of 28 healthy adult individuals of mean age (±SD) 45 ± 12 years including, 14 females, was collected in Vacutainer tubes with K_3_EDTA (BD, Franklin Lakes, NJ, USA), 7 samples per blood group. None of the individuals had been hospitalized in the last two years, had not taken antibiotics, and had not undergone dental interventions in the last six months. The blood samples were divided into two parts. One part for culturing (0.3 mL) and another part for direct DNA isolation (3 mL). All blood samples were tested for sterility by growing on Sabouraud and blood agar (BB-NCIPD, Sofia, Bulgaria).

We applied a modified resuscitation strategy previously developed by Emil Kalfin [10]. Three hundred microliters of blood sample were added to 2.7 mL of culture medium (1:10 *vol/vol*). Culturing was performed in sterile 10 mL polypropylene Falcon tubes (Corning Inc, Corning, NY, USA). The culture base medium was composed of Brain Heart Infusion (BHI), (BD, Franklin Lakes, NJ, USA) medium and 0.2% yeastolate (BD, Franklin Lakes, NJ, USA) adjusted at pH 6.8 and sterilized. Sterile (D+) sucrose at 10% final concentration and water-soluble form of vitamin K_3_-menadione sodium bisulfite (Sigma-Aldrich, Burlington, VT, USA) in a concentration of 1 mg/mL sterilized by filtration was added to the base medium. Resuscitation growth was induced at 43 °C for 24 h. Isolated blood microbiota was confirmed by Gram staining and 16S rRNA genes PCR analysis with universal primers [14].

### 2.2. DNA Isolation and Sequencing Analysis

After 24 h of culturing at 43 °C, the tubes were centrifuged at 3000 g for 20 min. The dark brown pellet was washed by vortexing with 10 mL of DNase I 1 U/mL (Thermo Fisher Scientific, Waltham, MA, USA) in ddH_2_O and incubated for 30 min at 37 °C to eliminate residual DNA contaminations associated with lysed nuclear blood cells, DNA from the culture medium or DNA traces in water. Centrifugation and washings were repeated three times to lyse the blood cells, to wash away medium and human blood cells DNA/RNA. The cell pellet was resuspended in 1 mL of lysis buffer (500 mM NaCl, 50 mM Tris-HCl, pH 8.0, 50 mM EDTA, and 4% sodium dodecyl sulfate). The same procedure was applied for direct microbiota isolation from whole blood. Briefly, 3 mL of blood were lysed with 10 mL of DNase I 1 U/mL in ddH_2_O and incubated for 30 min at 37 °C. After centrifugation at 3000 g for 20 min, the same procedure was repeated twice. The pellet from the lysed whole blood was resuspended in 1 mL of lysis buffer (500 mM NaCl, 50 mM Tris-HCl, pH 8.0, 50 mM EDTA, and 4% sodium dodecyl sulfate). The cell suspension was subjected three times to freeze-thawing in liquid nitrogen and a dry bath at 96 °C for 10 min, combined with vigorous vortexing or optional homogenization with silica/zirconium 0.1 mm beads (Biospec Products, Bartlesville, OK, USA) on a bead beater (Benchmark Scientific, Sayreville, NJ, USA) for 3 min at 4000 rpm. DNA isolation was performed according to the previously described procedure [27]. The extracted DNA was resuspended in 100 µL of sterile DNA/RNA free dH_2_O (Sigma-Aldrich, Burlington, VT, USA) DNA typical yield was >150 ng/µL with a 260/280 nm ratio of ~1.5–1.7.

For DNA analysis we applied 16S rRNA genes and ITS2 targeted sequencing on Illumina MiSeq (Illumina Inc., San Diego, CA, USA). DNA sequencing was performed at IMGM Laboratories GmbH (IMGM Laboratories GmbH, Martinsried, Germany). For bacteria identification V3–V4 hypervariable regions of the 16S rRNA genes were amplified with universal primers 314F and 805R producing amplicons of 465 bp. For fungi identification, ITS2 region was amplified with universal primers ITS3 and ITS4 producing amplicons of 400 bp. Bidirectional sequencing was performed. Both reads have a length of 300 bases, 2 × 300 bp paired-end (PE) reads.

### 2.3. Bioinformatics

The Illumina software MiSeq Reporter version 2.5.1.3 (Illumina Inc., San Diego, CA, USA) and the Illumina Sequence Analysis Viewer version 2.1.8 (Illumina Inc., San Diego, CA, USA) were used for imaging and evaluation of the sequencing run performance. Primary data analysis and QC, including signal processing and de-multiplexing, were performed using the MiSeq Reporter and the CLC Genomics Workbench version 9.5.3 (QIAGEN, København, Denmark). QIAGEN CLC Microbial Genomics Module was used for OTU taxonomical profiling according to manufacturer’s guidelines (https://digitalinsights.qiagen.com; accessed on 10 March 2021). As a first step read 1 and read 2 sequences from every sample were merged using the overlapping sequence information and taking the minimum overlap, potential mismatches, gaps and unaligned ends into account. The two reads were merged including overlaps of minimum of 20 bp without any mismatch and maximum unaligned end mismatches of 2 bp. Trimming of reads was performed according to TS primer sequences, base quality and read length, where a probability quality limit of 0.05 was applied to ensure high-quality data for subsequent analysis. To guarantee similarity and a sufficiently high level of sequence information for phylogenetic classification, sequences <320 bp were discarded for 16S analysis and longer sequences were trimmed down to this length. For ITS analysis, a fixed-length trimming of 150 bp was applied to the reversed sequences to make sure that the highly variable ITS2 region was kept for further analysis, while the homologous 5.8 rRNA gene region was discarded. Chimeric sequences, representing PCR and sequencing artifacts, were filtered out and discarded during this step. Out of each cluster, one reference sequence of an OTU was defined and is represented with one line each in the full result table. The obtained sequences were clustered into Operational Taxonomic Units (OTUs). OTUs were analyzed for sequence similarities against reference sequence databases-UNITE for ITS2 (https://unite.ut.ee/; accessed on 10 March 2021) and Greengenes for 16S rDNA (https://greengenes.secondgenome.com; accessed on 10 March 2021). For de novo OTUs with the highest combined abundance an additional BLAST search was applied to identify bacterial and fungal species that might have been missed in Greengenes and UNITE databases. The Standalone BLAST Setup for Windows was used (https://ftp.ncbi.nlm.nih.gov/blast/executables/LATEST/ncbi-blast-2.11.0+-x64-win64.tar.gz; accessed on 12 March 2021) comparing query sequences from de novo OTUs detected with the nucleotide collection database at NCBI considering all organisms. All OTUs (reads > 0) present in reagent controls, dH2O, and culture medium were subtracted from experimental samples to eliminate contaminating OTUs. For taxonomic affiliation, the interference created by the MiSeq sequencer and between samples was also considered. Raw data files and metadata can be accessed at Git Hub repository (https://github.com/yhodzhev/blood_microbiota; accessed on 12 May 2021).

### 2.4. Statistics

R version 3.6.0 was used for statistical analysis to compare between cultured and non-cultured group counts. These values for non-cultured and cultured groups of bacteria and fungi were compared with a t-test for independent samples. Significance was defined as a two-tailed *p*-value (*p* < 0.05) including control for false discovery rate (FDR). The average taxa content for each sample was calculated and expressed as means ± SD (Table 1). Analysis and visualization of microbiome data were performed on the Microbiome Analyst web-based service (https://www.microbiomeanalyst.ca; accessed on 8 April 2021). Its Marker data profiling section provides a graphical user interface for online application of the Microbiome Analyst R package for R version above 3.5.1 (https://github.com/xia-lab/MicrobiomeAnalystR; accessed on 8 April 2021) and provides numerous pipelines for metagenome analysis [28,29]. OTU abundance tables and corresponding metadata were uploaded on the server for further processing. OTU abundance was determined as a number of reeds without further transformation and normalization. Hierarchical clustering used the Euclidean distance and Ward algorithm. Heatmap visualization encoded the number of reads magnitude with color intensity. For statistical evaluation of the beta diversity analysis, a permutation ANOVA (PERMANOVA) was used. A between-sample comparison was performed via principal coordinate analysis (PCoA), ref. [30] including the Jensen–Shannon Divergence method [31] for determining distance and permutation ANOVA (PERMANOVA) for statistical validation. Two-dimensional PCoA plots were generated for further comparison of cultured and non-cultured samples by computation of the first two principal components, which then served as coordinates of those plots. Heatmap visualization and PCoA were performed with Microbiome Analyst software.

## 3. Results and Discussion

We induced resuscitation of latent/dormant microbial cells in blood samples of healthy individuals applying temperature and chemical stress conditions. Cultivation was for 24 h at 43 °C. The explosive growth of microbial structures was observed by light microscopy on Gram-stained slides. Cultured and non-cultured blood samples from healthy individuals were analyzed by targeted sequencing of 16S rRNA and ITS markers. The sequencing yield was 11.22 Gb with filter-passed clusters of 18.7 Mio (82.7%) and Q30 > 70%. The total number of identified bacterial OTUs were 3890 comprising 829 984 reads. For fungi, we obtained 34,980 OTUs comprising 3,046,992 reads. We established a filtering procedure to validate an OTU that included the elimination of all OTUs associated with contaminations by DNAs in reagents, culture medium, and negative controls. We eliminated all OTUs with less than 100 reads of combined abundance among all samples. The number of reads established as a threshold for a taxonomic assignment was set up at >10 reads per OTU. The total number of valid OTUs for bacteria identified in Greengenes, UNITE, and GenBank databases was 241, and 272 for fungi. Based on identified OTUs, we calculated the average number of bacterial and fungal taxa per sample (Table 1). The number of unidentified microbial taxa in bacteria was 13%, and 17.2% in fungi. This is due to limitations of the applied 16S and ITS markers and the available databases which cover only currently identified taxa.

The number of identified bacterial taxa per sample was almost twice more in cultured samples compared to non-cultured ones. Each blood sample had individually specific microbial biodiversity independent of sex or blood type [12].

Most bacterial taxa were enriched by the applied resuscitation strategy according to the OTUs combined abundancy. For the majority of the samples, the numbers of bacterial taxa after culture enrichment were twice higher per sample. For fungal taxa, we identified that the proposed resuscitation procedure is less appropriate, and most taxa do not grow well, but a significant number of unique taxa not identified in non-cultured samples were revealed. Combining the results of all non-cultured and cultured samples, we identified rich microbial biodiversity in the blood of healthy individuals (Table 2).

In our set of cultured and non-cultured samples, we identified 513 bacterial and fungal OTUs. The bacterial biodiversity among samples included 241 OTUs distributed in 40 orders containing 66 families and 105 genera. Fungal biodiversity accounted for 272 OTUs distributed in 61 orders, 105 families, and 133 genera. A significant number of taxa were unique in non-cultured or cultured samples. Species belonging to 5 bacterial and 54 fungi genera were non-culturable within the tested medium and temperature conditions.

Dominant bacterial phyla among non-cultured samples were Proteobacteria 92.97%, Firmicutes 2.18%, Actinobacteria 1.74% and Planctomycetes 1.55%, while among cultured samples Proteobacteria were 47.83%, Firmicutes 25.85%, Actinobacteria 16.42%, Bacteroidetes 3.48%, Cyanobacteria 2.74%, and Fusobacteria 1.53%. The fungi phyla Basidiomycota, Ascomycota, and unidentified fungi were 65.08%, 17.72%, and 17.2% respectively among the non-cultured samples, while among the cultured samples they were 58.08%, 21.72%, and 20.2% respectively.

Bacterial orders that remained non-culturable, compared to blood microbiota isolated from fresh blood collection, were Sphingomonadales, Rhizobiales, and Rhodospirillales. Species of bacterial orders Bacillales, Lactobacillales, and Corynebacteriales showed the best cultivability. Among the fungi orders Tremellales, Filobasidiales and Russullales were identified as non-culturable. Species of fungi orders Pleosporales, Saccharomycetales, and Helotiales were among the best culturable. For many microbial families and genera, we identified that OTU reads of cultivated samples are dozens of or even hundreds of times more abundant compared to non-cultured ones, i.e., 135 reads of genus *Micrococcus* in non-cultured samples versus 3656 reads in cultured samples (Appendix A).

According to our estimations, microbial quantity in the blood of healthy individuals is relatively high. By light microscopy, blood microbiota in non-cultured samples could not be observed. The size of the elementary bodies, composing the blood microbiota, by electron microscopy was estimated at 150–200 nm [8,10,12,16,26]. If we consider that a bacterium of 4.5 Mbp has a mass of 4.62 fg and that from 3 mL of fresh blood we extract about 5–7 µg of not highly purified microbial DNA, then the rough estimation of microbial structures in blood per mL will be high. On Gram-stained preparations of 24 h cultured blood samples, we observe 30–150 microbial cells per field (data not shown).

Data in Table 1 demonstrates that the combination of stress factors as high concentration of vitamin K and cultivation at 43 °C induced survival mechanisms through explosive microbial growth. Most microbial species grow well in tested nutrient medium and culture conditions. This is evident by the increased number of OTU reads after cultivation. Cultured samples demonstrated a significant number of taxa not detected in non-cultured ones. Thus, this proves the need when describing the full biodiversity to test pairs of cultured and non-cultured samples. Significant hidden biodiversity could be displayed in cultured samples. In total, 55 bacterial and 39 fungal genera were unique.

Representatives of non-culturable bacteria in blood according to the reads abundance are members of orders Sphingomonadales, Rhizobiales, and Rhodospirillales (Figure 1). The proposed resuscitation strategy of blood microbiota was more appropriate for bacteria than for fungi (Table 2). Tested culture medium and temperature stress conditions are not appropriate for growing representatives of fungi orders Tremellales, Polyporales, Filobasidiales, some unidentified Basidiomycota and Agaricomycetes, Russulales, and Sporidiobolales (Figure 2). Analysis was performed on fungi identified in humans, animals, and Insecta.

We examined in detail the families with the highest difference in relative reads abundancy between cultured and non-cultured microbiota. Heatmap and clustering analysis confirmed diversity in cultivability for both bacteria (Figure 3a) and fungi (Figure 3b). At the family level, bacteria showed the distinct formation of two clusters linked with the experimental conditions. Fungi exemplified families with minimal fluctuation in reads in cultured and non-cultured samples.

Principal coordinates analysis gave additional support to heatmap findings (Figure 4). We demonstrated further statistical validation of the intergroup difference as revealed by the PCoA. Analysis showed that bacterial taxa vary significantly in sample abundance. Figure 4b showed significant differences in beta-diversity between non-cultured and cultured bacteria (PERMANOVA, F = 7.43; R^2^ = 0.12; *p* < 0.001), while fungi indicated (Figure 4d) similar beta-diversity between non-cultured and cultured groups (PERMANOVA F = 0.81; R^2^ = 0.014; *p* = 0.59).

Combining the OTUs from all cultured and non-cultured samples, we identified 105 bacterial genera. Similar results were reported for samples from the upper female reproductive tract considered to be sterile [32]. At the phylum level, we identified that the blood microbiome was predominated by Proteobacteria 93% followed by Firmicutes 2.18%, Actinobacteria 1.74%, and Planctomycetes 1.55%. These findings mirror previous studies [14,15,33,34,35] and further support the concept of a core blood microbiota dominated by several key phyla. Although Planctomycetes are considered marine and environmental microorganisms, recently they have been confirmed as common in the human gut [36]. In seven types of cancer, intracellular microbial structures belonging to over 500 bacterial species have been identified in the affected parts and the adjacent healthy tissue of the “sterile” internal organs [6]. The intratumor bacteria were mostly intracellular and detected in all clinical samples. In a recent study, Japanese authors demonstrated specific microbial species in blood from patients with liver cirrhosis and control healthy individuals. In the studied blood samples, the authors described in cancer tissues the presence of OTUs belonging to 183 bacterial genera, while in the control group of healthy individuals they proved 123 bacterial genera [37].

Our results demonstrate that blood microbiota is a viable ecological niche. Blood samples cultured under stress conditions exhibit hidden microbial diversity, which in uncultured samples may remain non-detected even with metagenomic analysis. By culturing the blood microbiota, the existence of which is questioned by some authors and considered as DNAemia of cell-free circulating DNA [19,20], we identified well culturable bacteria and fungi as normal blood microbiota in healthy individuals. We showed that blood DNAemia in healthy individuals is associated with the presence of culturable microbiota in the blood.

A large-scale study organized by the Human Microbiome Consortium demonstrated in 18 anatomical sites in the human body the presence of 5177 suspected bacterial species (taxonomic profiles), of which only 800 were cultured [3].

An open question is: how do microbes enter the bloodstream? Culturally, bacteremia has been demonstrated after tooth extraction, tooth brushing, or applying oral irrigation [38,39]. It is assumed that the presence of transient microflora in the circulatory system is well tolerated by healthy individuals. The hypothesis that the origin of bacteria in the blood is a consequence of gastrointestinal microbial translocation is well accepted [14]. The main source of microorganisms in the blood is the intestinal and oral microbiota by atopobiosis, i.e., translocation of microbial cells in the blood or other tissues [17]. Atopobiosis can contribute to the development of chronic non-communicable inflammatory diseases. Blood microbiota persists in a latent/dormant state. Our knowledge of the latency, dormancy, and persistency of microbial species is very limited.

## 4. Conclusions

Numerous studies have proven the presence of microbial species in the tissues of internal organs, blood, and body fluids in sick and healthy people. The role of the blood microbiome in human health is an open field for innovative research with the potential to become a new medical discipline. It can be assumed that in the coming years, new research teams will focus on studying the blood microbiome and its relation to many diseases of presumed infectious etiology, such as arthritis [13], sarcoidosis [40], blood anemia [17], latent tuberculosis [41], and many others [7,17].

## Figures and Tables

**Figure 1 microorganisms-09-01464-f001:**
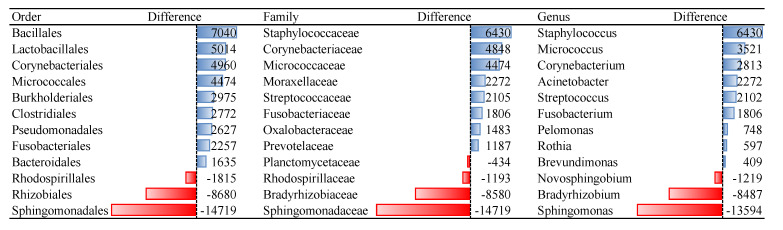
Differences of reads abundancy in identified bacterial taxa among non-cultured and cultured blood samples. Effect of the culture medium and temperature conditions on the blood microbial resuscitation assessed according to the number of OTU reads.

**Figure 2 microorganisms-09-01464-f002:**
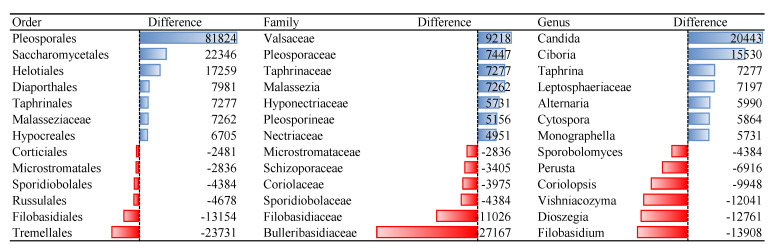
Differences of reads abundancy in identified fungal taxa among non-cultured and cultured blood samples. Effect of the culture medium and temperature conditions on the blood microbial resuscitation assessed according to the number of OTU reads.

**Figure 3 microorganisms-09-01464-f003:**
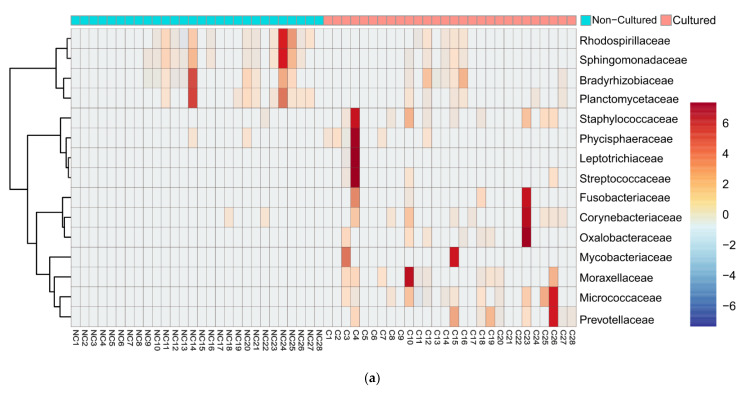
Heatmap diagram of bacterial and fungal blood microbiota composition at family level of cultured and non-cultured samples for each individual, (**a**) 15 bacterial families identified with highest abundancy difference between the two groups for each individual, (**b**) 15 fungi families with highest abundancy difference for each individual. Non-cultured (NC) and cultured (C) blood samples of individuals from 1 to 28.

**Figure 4 microorganisms-09-01464-f004:**
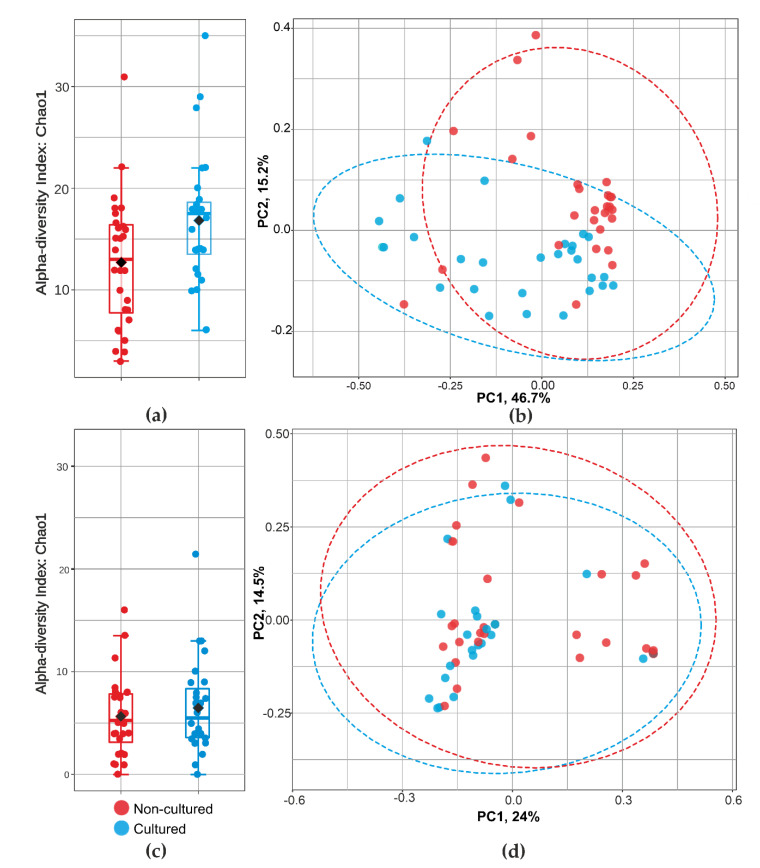
Community diversity analysis for cultured and non-cultured blood microbiota at family level for bacteria and fungi. alfa-diversity (Chao 1 index) measuring bacterial (**a**) and fungal (**c**) richness. Graphics (**b**,**d**) show bacterial and fungal beta-diversity distance between samples in two dimensions, with individual points representing OTU distribution of each sample based on the principal coordinates analysis (PCoA).

**Table 1 microorganisms-09-01464-t001:** Number of identified bacterial and fungal taxa per sample for cultured and non-cultured samples. Mean ± SD.

	BACTERIA		FUNGI	
Taxon	Non-Cultured	Cultured	Non-Cultured	Cultured
Order	5.29 ± 3.02	11.61 ± 6.13 (***)	6.64 ± 3.46	6.04 ± 3.45 (ns)
Family	6.64 ± 4.75	13.18 ± 7.74 (***)	7.71 ± 4.18	7.25 ± 4.06 (ns)
Genus	7.68 ± 5.58	14.86 ± 9.78 (***)	7.93 ± 4.58	7.36 ± 3.94 (ns)

SD-standard deviation; significance level from independent t-test *p* < 0.001 (***); (ns)-non significant.

**Table 2 microorganisms-09-01464-t002:** Number of identified taxa among non-cultured and cultured blood samples.

	BACTERIA						FUNGI					
	Orders		Families		Genera		Orders		Families		Genera	
	NC	C	NC	C	NC	C	NC	C	NC	C	NC	C
Common	24	24	40	40	45	45	25	25	38	38	40	40
Unique	0	16	0	26	5	55	19	17	37	30	54	39
Total	24	40	40	66	50	100	44	42	75	68	94	79

C-cultured sample; NC-non-cultured sample.

## Data Availability

The raw data files, as well as metadata, can be accessed freely at Git Hub repository at the following web address: https://github.com/yhodzhev/blood_microbiota; accessed on 12 May 2021.

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
