# Peer review of "Culturable and Non-Culturable Blood Microbiota of Healthy Individuals"

_microorganisms, 2021, doi:10.3390/microorganisms9071464_

Round 1

Reviewer 1 Report

The work by Panaiotov et al in which they explore the culturable and non-culturable microbiota in the circulating blood of healthy individuals is an interesting read. However, there are some issues that need to be addressed before the publication of this article.

Line 97: What are the mean (standard deviation) age group and gender distribution of these healthy individuals?

Section 2.3: There is no mention of the pipeline that was used to process the raw reads. Additionally, the authors miss out on the richness and diversity analyses. Authors can look at these representative articles (https://doi.org/10.2215/CJN.12161018 and https://dx.doi.org/10.1097%2FMEG.0000000000001494).

Line 152: reeds → reads

Section 2.4: Again, no reference to any of the software or packages used for the statistical analyses.

Line 184: This should be discussed as why the number was more in cultured samples as compared to non-cultured one.

Table 2: The authors provide the statistics on taxa identified in cultured and non-cultured blood samples. However, it will be of high interest if additional details are provided.

e.g. There are 45 bacterial genera that are common between these two types of blood samples. However, it is not clear if they show similar abundance patterns across these samples.

Similarly, the authors identify 5 and 55 unique bacterial genera in NC and C samples, respectively. It should be indicated which taxonomic groups (at the order, family, or genus) are dominant if any in these two different samples. These data could be provided as tables or figures.

Author Response

We thank you for the critical comments on our manuscript.  

Reviwer 1: Line 97: What are the mean (standard deviation) age group and gender distribution of these healthy individuals?

Answer: Corrections were introduced. Lane 97 was modified: Blood of 28 healthy adult individuals {females : males (1:1)}  was collected in Vacutainer tubes with K3EDTA (BD, USA), 7 samples per blood group. None of the individuals had been hospitalized in the last two years, had not taken antibiotics, and had not undergone dental interventions in the last six months.

Reviwer 1: Section 2.3: There is no mention of the pipeline that was used to process the raw reads. Additionally, the authors miss out on the richness and diversity analyses. Authors can look at these representative articles (https://doi.org/10.2215/CJN.12161018 and https://dx.doi.org/10.1097%2FMEG.0000000000001494).

Answer: A major revision of section 2.3. Bioinformatics and 2.4. Statistics was made. Detailed description of the pipeline appied to process the raw reads was given. The following text was added:

‘The Illumina software MiSeq Reporter version 2.5.1.3 (Illumina Inc., USA) and the Illumina Sequence Analysis Viewer version 2.1.8 (Illumina Inc., USA) were used for imaging and evaluation of the sequencing run performance. Primary data analysis and QC, including signal processing and de-multiplexing, was performed using the MiSeq Reporter and the CLC Genomics Workbench version 9.5.3 (QIAGEN, Denmark). QIAGEN CLC Microbial Genomics Module was used for OTU taxonomical profiling according to manufacturer’s guidelines (https://digitalinsights.qiagen.com ). As a first step read 1 and read 2 sequences from every sample were merged using the overlapping sequence information and taking the minimum overlap, potential mismatches, gaps and unaligned ends into account. The two reads were merged including overlaps of minimum 20 bp without any mismatch and maximum unaligned end mismatches of 2 bp. Trimming of reads was performed according to TS primer sequences, base quality and read length, where by a probability quality limit of 0.05 was applied to ensure high quality data for subsequent analysis. To guarantee similarity and a sufficiently high level of sequence information for phylogenetic classification, sequences < 320 bp were discarded for 16S analysis and longer sequences were trimmed down to this length. For ITS analysis, a fixed length trimming of 150 bp was applied to the reversed sequences to make sure that the highly variable ITS2 region was kept for further analysis, while the homologous 5.8 rRNA gene region was discarded. Chimeric sequences, representing PCR and sequencing artefacts, were filtered out and discarded during this step. Out of each cluster one reference sequence of an OTU was defined and is represented with one line each in the full result table. The obtained sequences were clustered into Operational Taxonomic Units (OTUs). OTUs were analyzed for sequence similarities against reference sequence databases - UNITE for ITS2 (https://unite.ut.ee/) and Greengenes for 16S rDNA (https://greengenes.secondgenome.com). For de novo OTUs with the highest combined abundance an additional BLAST search was applied to identify bacterial and fungal species that might have been missed in Greengenes and UNITE databases. The Standalone BLAST Setup for Windows was used (https://ftp.ncbi.nlm.nih.gov/blast/executables/LATEST/ncbi-blast-2.11.0+-x64-win64.tar.gz) comparing query sequences from de novo OTUs detected with the nucleotide collection database at NCBI considering all organisms. All OTUs (reads > 0) present in reagent controls, dH2O, and culture medium were subtracted from experimental samples to eliminate contaminating OTUs. For taxonomic affiliation, the interference created by the MiSeq sequencer and between samples was also considered. Raw data files and metadata can be accessed at Git Hub repository, https://github.com/yhodzhev/blood_microbiota.’

Reviwer 1: Additionally, the authors miss out on the richness and diversity analyses.

Answer: In order to improve the visualization of the results we added to Figure 4 additinal graphics for bacterial and fungal richness presenting Chao1 index for alfa diversity in samples. Beta diversity is presented with PCoA graphics.  

Reviwer 1: Line 152: reeds → reads –  

Answer: Correction added.

Reviwer 1: Section 2.4: Again, no reference to any of the software or packages used for the statistical analyses.

Answer: The following improvements were added in section 2.4. Statistics:

2.4. Statistics

R version 3.6.0 was used for statistical analysis to compare between cultured and non-cultured group counts. These values for non-cultured and cultured groups of bacteria and fungi were compared with t-test for independent samples. Significance was defined as two-tailed P value (P < 0.05) including control for false discovery rate (FDR). The average taxa content for each sample was calculated and expressed as means ± SD (Table 1). Analysis and visualization of microbiome data were performed on the Microbiome Analyst web-based service (https://www.microbiomeanalyst.ca). Its Marker data profiling section provides graphical user interface for online application of the Microbiome Analyst R package for R version above 3.5.1 (https://github.com/xia-lab/MicrobiomeAnalystR) and provides numerous pipelines for metagenome analysis [28,29]. OTU abundance tables and corresponding metadata were uploaded on the server for further processing. OTU abundance was determined as a number of reeds without further transformation and normalization. Hierarchical clustering used the Euclidean distance and Ward algorithm. Heatmap visualization encoded the number of reads magnitude with color intensity. For statistical evaluation of the beta diversity analysis a permutation ANOVA (PERMANOVA) was used.’

Reviewer 1: Line 184: This should be discussed as why the number was more in cultured samples as compared to non-cultured one.

Answer: This is an empirical fact. We could only speculate that it is associated with the experimental design (culture medium, temperature, high saccarose content and vitamin K).

Table 2: The authors provide the statistics on taxa identified in cultured and non-cultured blood samples. However, it will be of high interest if additional details are provided.

e.g. There are 45 bacterial genera that are common between these two types of blood samples. However, it is not clear if they show similar abundance patterns across these samples.

Similarly, the authors identify 5 and 55 unique bacterial genera in NC and C samples, respectively. It should be indicated which taxonomic groups (at the order, family, or genus) are dominant if any in these two different samples. These data could be provided as tables or figures.

Answer: We consider that Figures 1 and 2 give sufficient detailed information which bacterial and fungi orders, families and genera among cultured and non-cultured samples have unique characteristics – grow well or do not grow. In our discussion at several places we underline that ‘Representatives of non-culturable bacteria in blood according to the read abundance are members of orders Sphingomonadales, Rhizobiales, and Rhodospirillales (Figure 1). The proposed resuscitation strategy of blood microbiota was more appropriate for bacteria than for fungi (Table 2). Tested culture medium and temperature stress conditions are not appropriate for growing representatives of fungi orders Tremellales, Polyporales, Filobasidiales, some unidentified Basidiomycota and Agaricomycetes, Russulales, and Sporidiobolales (Figure 2).’ And For many microbial families and genera, we identified that OTU reads of cultivated samples are dozens of or even hundreds of times more abundant compared to non-cultured ones, i.e. 135 reads of genus Micrococcus in non-cultured samples versus 3 656 reads in cultured samples (Table S1). Detailed information about the characteristics of the samples could be extracted from Supplementary Table S1.

Reviewer 2 Report

This work discusses the culture bacterial microbiota on healthy people, through NGS analysis. Understanding the microbiome, its complexity and role in health and diseases has proven to be crucial to future therapeutic approaches and response to drugs.

The major issue this reviewer finds is that the difference of culturable and non-culturable fungal species – which is an important find - needs to be netter explained in the discussion section.

Other points are:

Abstract:

  • This reviewer thinks the abstract should be reduced. It is extremely long. Please summarize this section;
  • “...while others have had isolated a limited number of bacterial species by culture.” – fungal species have also been isolated. Please correct this;
  • Please define abbreviations, before using them;

Introduction:

- These recent works would enrich the MS: doi: 10.1080/10408398.2020.1760202;  doi: 10.1016/j.rceng.2019.07.018.

M&M:

  • Brand and country of manufacturers need to be places in every reagent and material (e.g. Sabouraud). Please check the entire MS and adjust;
  • Statistical software used?

Results and Discussion:

  • Table 1 and 2: correct (u)necessary bold;
  • Cultured and uncultured fungal species have similar values. This is important and needs to be better and deeper explored in the Discussion;
  • Orders do not need to be in italic form. Only genera and species. Please correct this;
  • All figures should be placed in “Results” section;
  • Figure 3 and 4 need to be sharper to be better understood.

Author Response

We thank you for the critical comments on our manuscript.

Reviwer 2: The major issue this reviewer finds is that the difference of culturable and non-culturable fungal species – which is an important find - needs to be netter explained in the discussion section.

Answer: This is an empirical fact. Tables and Figures the empirical data. We could only speculate that the difference of culturable and non-culturable fungal species is associated with the experimental design (culture medium, temperature, high saccarose content and vitamin K). On the bases of the empirical data we made general conlusions such as ‘Among the fungi orders Tremellales, Filobasidiales and Russullales were identified as non-culturable. Species of fungi orders Pleosporales, Saccharomycetales, and Helotiales were among the best culturable.’

Reviewer 2: This reviewer thinks the abstract should be reduced. It is extremely long. Please summarize this section;

Answer: The abstract section from 398 words was reduced to 321 words.

Reviewer 2: “...while others have had isolated a limited number of bacterial species by culture.” – fungal species have also been isolated. Please correct this;

Answer: Correction added – and fungal.

Reviewer 2: Please define abbreviations, before using them;

Answer: Corrections were done. The following abbreviations were defined:

  • internal transcribed spacer (ITS)
  • operational taxonomic units (OTU)

Reviewer 2: Brand and country of manufacturers need to be places in every reagent and material (e.g. Sabouraud). Please check the entire MS and adjust;

Answer: Corrections were introduced. All necessary clarifications  for manufacturers of reagents and equipment  were included.

Reviewer 2: Statistical software used?

Answer: Corrections added. Name and access address of all used softwares were added where missing.

Revewer 2: Table 1 and 2: correct (u)necessary bold;

Answer: Corrections were introduced.

Reviewer 2: Cultured and uncultured fungal species have similar values. This is important and needs to be better and deeper explored in the Discussion;

Answer: Figure 3 and Figure 4 support the finding that most of cultured and uncultured fungal species have similar values. Diversity of fungal taxa among cultured and non-cultured blood samples is an open question and is difficult to give a concrete answer why they have similar values, although samples demonstrated rich fungal biodiversity. In the text we explain: For fungal taxa, we identified that the proposed resuscitation procedure is less appropriate and most taxa do not grow well, but a significant number of unique taxa not identified in non-cultured samples were revealed.’ Further we clarified that: ‘The fungi phyla Basidiomycota, Ascomycota, and unidentified fungi were 65.08%, 17.72% and 17.2% respectively among the non-cultured samples, while among the cultured samples they were 58.08%, 21.72% and 20.2% respectively.And:Among the fungi orders Tremellales, Filobasidiales and Russullales were identified as non-culturable. Species of fungi orders Pleosporales, Saccharomycetales, and Helotiales were among the best culturable.   Heatmap diagrams demonstrate that Fungi exemplified families with minimal fluctuation in reads in cultured and non-cultured samples.’ And applying Principal Coordinates Analysis we also conclude that as shown on Figure 4b fungi tended to form more disperse clusters (PERMANOVA, F = 0.81; R2 = 0.014; P < 0.59).

Reviewer 2: Orders do not need to be in italic form. Only genera and species. Please correct this;

Answer: Corrections were introduced. Throughout the text all phyla, orders and family level assignments were corrected in non-italic form.

Reviewer 2: All figures should be placed in “Results” section;

Answer: In order to simplify and improve the logical presentation of the results and their discussion, we combined Results and Discussion sections in one.

Reveiwer 2: Figure 3 and 4 need to be sharper to be better understood.

Answer: Firgure 3 and 4 were substituted and prepared with more contrast and sharp color design.

Reviewer 3 Report

An interesting, novel study that proposes an intriguing hypothesis. The study is of scientific merit and the methods presented are clearly stated. 

As this is a novel area of research the authors should continue their studies also for patients with systemic diseases or conditions.

It was not specified in the materials and methods section weather the patients underwent antibiotic treatment in the last months.

Author Response

We thank you for the critical comments on our manuscript.

Reviewer 3: It was not specified in the materials and methods section weather the patients underwent antibiotic treatment in the last months.

Answer: In section Materials and Methods the following clarification was added: ‘Blood of 28 healthy adult individuals of mean age (± SD) 45 ± 12 years, including 14 females, was collected in Vacutainer tubes with K3EDTA (BD, USA), 7 samples per blood group. None of the individuals had been hospitalized in the last two years, had not taken antibiotics, and had not undergone dental interventions in the last six months.’  

Round 2

Reviewer 2 Report

Thank you for the adjustments and corrections.

Author Response

The authors would like to thank this reviewer for his/her valuable time taken to evaluate the manuscript, and for pointing out improvement possibilities and critical notes. We tried to answer the questions comprehensively.

  1. The name is confusing.

The terms "Culturable” and “Non-culturable” are not clear.

Do you mean Viable but not culturable - defined as live bacteria, but which do not either grow or divide? If so, it has to be stated clearly throughout the manuscript.  

Answer: In our experimental model we test one specific culture medium designed to resuscitate microbiota in cultured and non-cultured blood samples. As demonstrated by our results we detected well culturable, less culturable, and non-culturable microbiota (Figure 1 and Figure 2) at order, family, or genus level. In conclusion, we quantified the capacity of a specific medium applied for culturing of blood microbiota in healthy individuals. Other culturing conditions and media should be tested for optimization and better characterization of blood microbiota in healthy and diseased individuals.

  1. It is necessary to state how can you separate transition bacteremia vs microbiota?

Answer: In our investigations, we studied the blood microbiota in healthy individuals. Bacteremia is uncommon in healthy subjects. As it is stated in the manuscript, “None of the individuals had been hospitalized in the last two years, had not taken antibiotics, and had not undergone dental interventions in the last six months.”

  1. The introduction is messy. Please add, along (or instead) with the description of papers which were published 50 years ago, some recent studies for the blood microbiome based on the sequence techniques. The outcomes from these studies have to be discussed.  

Answer: In the introduction, we gave a historical overview of how studies on blood microbiota started and continued to evolve. One of the first studies was published in Nature in 1969. The most recent cited publication in the introduction is from 2020 in Science. We underlined, that the dilemma for the blood microbiota divided researchers into three groups. “The first one supports the hypothesis, that blood is not as sterile as previously supposed and that blood microbiota is naturally existing in the blood of healthy individuals [8–15]. Dormant, latent, or non-culturable microbial forms should be considered [16,17]. The second assumes that identified microbial DNAs in the blood are cell-free circulating DNAs [18–20]. The third one denies the existence of enigmatic blood microbiota in healthy individuals [21,22].” The outcomes from these studies are discussed throughout the text. 

  1. The authors describe the cultivation of all microorganisms on a medium based on a Brain Heart Infusion for 24h. However, Bradyrhizobium spp, normally  grow on a medium that contains only salts, MES/HEPES buffer, pH 6·7.5, and arabinose and gluconate as carbon sources. And they do not grow so quickly. The same is true for many other bacteria and fungi which were underrepresented and named as “non-culturable” in this article. If the authors suggest that since these microorganisms are found in blood, they should grow on BHI, this statement is not true, since these bacteria might simply pass the bloodstream reflecting a transition bacteremia

Answer: For resuscitation of blood microbiota we tested a culture medium composed of Brain Heart Infusion medium and 0.2% yeastolate adjusted at pH 6.8 and sterilized. Sterile (D+) sucrose at 10% final concentration and water-soluble form of vitamin K3-menadione sodium bisulfite in a concentration of 1 mg/mL sterilized by filtration was added to the base medium. Resuscitation growth was induced at 43°C for 24 h. We quantitatively measured the culturable part of the blood microbiota of healthy individuals. We planned our study assuming that bacteriemia in healthy individuals is rare and uncommon.

  1. Lines 92-93. The statement "This study fills in a knowledge gap and provides an analysis of the cultivability of blood circulating microbiota" is inaccurate. You use only one specific medium applied for culturing of blood microbiota. This blood medium is not optimal for many bacteria and fungi that have not grown on it. Moreover, you use not optimal temperature conditions. So you how can you claim "cultivability" in a broad meaning of this term?

Answer: As previously explained we tested one specific medium for resuscitation of blood microbiota in healthy individuals. Data in Table 1 demonstrate that the combination of stress factors as high concentration of vitamin K and cultivation at 43°C induce survival mechanisms through explosive microbial growth.  We applied targeted sequencing of 16S rDNA and internal transcribed spacer (ITS) markers on cultured and non-cultured blood to study the biodiversity of blood microbiota. Our quantitative results demonstrated, that dominant bacterial phyla among non-cultured samples were Proteobacteria 92.97%, Firmicutes 2.18%, Actinobacteria 1.74% and Planctomycetes 1.55%, while among cultured samples Proteobacteria were 47.83%, Firmicutes 25.85%, Actinobacteria 16.42%, Bacteroidetes 3.48%, Cyanobacteria 2.74%, and Fusobacteria 1.53%. Fungi phyla Basidiomycota, Ascomycota, and unidentified fungi were 65.08%, 17.72%, and 17.2% respectively among non-cultured samples, while among cultured samples they were 58.08%, 21.72%, and 20.2% respectively.

  1. For DNA analysis we applied 16S rRNA genes and ITS2 targeted sequencing on Illumina MiSeq (Illumina Inc., USA). DNA sequencing was performed at IMGM Laboratories GmbH (Martinsried, Germany). For bacteria identification V3-V4 hypervariable regions of the 16S rRNA genes were amplified with universal primers 314F and 805R producing amplicons of 465 bp. – It is contractionary of how it is written.  What do you mean “For DNA analysis” vs “For bacteria identification”?

Answer: Microbial biodiversity was described by applying DNA NGS analysis combining with bioinformatics. Bacteria were identified by applying analysis of the reference hypervariable V3-V4 region of the 16S rRNA genes. ITS2 reference region was used for DNA sequencing and identification of fungi.

  1. Lines 101-102. What do you mean "7 samples per blood group"?

Do you mean blood type? English has to rechecked throughout the manuscript.

Answer: A blood type (also known as a blood group) is a classification of blood, based on the presence and absence of antibodies and inherited antigenic substances on the surface of red blood cells (RBCs). Both terms are used in the scientific literature.

  1. lines 235-236 "Elimination of all OTUs with <100 reads of combined abundance among all samples”  - The wording is incorrect. Please check the English.

Answer: The sentence was modified: Elimination of all OTUs with less than 100 reads of combined abundance among all samples.

  1. 254-256  The number of reads established as a threshold for a taxonomic assignment was set up at >10 reads per OUT – please add the reference citing this threshold? How is it relevant with your lines 235-236 “Elimination of all OTUs with <100 reads of combined abundance among all samples”?

Answer: Elimination of all OTUs with less than 100 reads of combined abundance among all samples and the number of >10 reads per OTU for a taxonomic assignment are stringent filtering criteria established by us in this study to validate an OTU.

  1. Lines 255-258  What do you mean under blood type in “Each blood sample had individually specific microbial biodiversity independently of sex or blood type.” Please provide the analysis for both of these analyses (sex and blood type).

Answer: This empiric result is confirmed by Figure 3 and Figure 4 where the profiles of each individual are presented in reference 12. Reference 12 was added to the manuscript.    

  1. Line 299-300 The size of blood microbiota by electron microscopy was estimated at 150-200 nm.  It is unclear what do you mean under “the size of microbiota”.

Answer: We mean that the size of the single microbial cell (elementary body) under electron microscopy was estimated at 150-200 nm [8,10,12,16,26]. We added in the text (elementary bodies) to clarify the meaning.

  1. Line 303-304 On Gram-stained preparations of 24 h cultured blood samples, we observe 30-150 microbial structures per field (data not shown). What do you mean under "microbial structures"?

Answer: We mean microbial structures stained by Gram and observed under light microscopy.

  1. Line 305 - Data on Table 1 demonstrate – should be demonstrates. Please check the English in the manuscript. You have multiple errors and typos

Answer: The text has been corrected.

  1. Line 306 - and cultivation at 43C induce – should be induced. Please check the English in the manuscript. You have multiple errors and typos

Answer: The text has been corrected.

  1. Lines 309 – 312. English has to be improved.

Answer: The text has been corrected.

  1. Line 312 "In total 55 bacterial and fungal genera were unique." – it is completely unclear to what data is it attributed? What do you mean under unique?

Answer: The meaning of ‚unique‘ in this case is that 55 bacterial and 39 fungal genera were identified only in cultured samples. Table 1 reflects the results at order, family, and genus levels.  The text has been improved. In total 55 bacterial and 39 fungal genera were unique.

  1. Figures 1 and 2. Are unclear You state that "Difference of reads abundancy in identified bacterial/fungal taxa among non-cultured and cultured blood samples. Effect of the culture medium and temperature conditions assessed according to the number of OTU reads."

Answer: Figure 1 and Figure 2 show the differeces in the number of reads identified among non-cultured and cultured samples identified for concrete order, family, and genus taxa. ‚Difference‘ was replaced with ‚Differences‘.

  1. Where are the data for “Effect of the culture medium and temperature conditions” it is unclear.

Answer: Throughout the text, we describe results obtained on cultured and non-cultured blood samples. We assessed the effect of the culture medium and temperature conditions. The quantitative assessment is expressed in differences in the number of OTU reads.  

  1. Figure 3a and 3b. It is unclear what S1-S56 stands for? Do I understand correctly that each blood samples was analyzed using “non-cultured” and “cultured” method? If so, how are these data reflected in the heatmaps?  How can one compare using “non-cultured” and “cultured” data from the same individual?  

Answer: On Figures 3a and 3b lanes from S1, S3, S5 …..etc. to S55 marked with blue color on top are non-cultured samples. Lanes from S2, S4, S6 …..etc. to S56 marked with pink color on top are cultured samples.  We compare the results of cultured and non-cultured samples. Our statistical analysis in Tables 1 and 2 and Figures 1-4 shows differences between cultured and non-cultured individual samples.  

  1. Lines 371-378 – How this part is relevant to your findings? Please consider rewriting or need to be removed.

Answer: Lines 371-378 cite similar results of other authors studying sterile body sites – internal organs and blood. Their results give support to our findings.

  1. Lines 384-386 – to broad statement. Needs to be limited to your study.

Answer: In our experiments, we eliminated the cell-free circulating DNA from the samples by treatment with DNase I. By culturing the blood microbiota we identified well culturable bacteria and fungi as normal blood microbiota in healthy individuals. We showed that blood DNAemia in healthy individuals is associated with the presence of culturable microbiota in the blood. Our results are consistent with the conclusions of the study.

  1. Lines 387-389 . How the phrase is related to the paragraphs before and after it?  

Answer: Aim of the Human Microbiome Project is to describe the full human microbiome. Till now, the data demonstrate the presence of 5177 suspected bacterial species of which only 800 were cultured. Our data also show that a significant part of blood microbiota is non-culturable, but indirectly detected by NGS analysis.

  1. Line 39 5- please replace microflora with microbiota. Microflora is an outdated and inaccurate term.

Answer: The term ‚microflora‘ was replaced with ‚microbiota‘.

  1. Line 396 – How the phrase "Atopobiosis can contribute to the development of chronic non-communicable inflammatory diseases” is relevant with the previous sentences?

Answer: It is well accepted hypothesis that the origin of bacteria in the blood is a consequence of gastrointestinal microbial translocation. Gastrointestinal dysbiosis may lead to chronic diseases. Atopobiosis of gastrointestinal microbiota in blood has recently been associated with chronic arthritis, autism, and even Alzheimer disease.

  1. Line 398: "Microbial cells persist in a latent/dormant state". What are you trying to say? What is written is that all microbial cells persist in a dormant state. English has to be improved.

Answer: This is essential to clarify. We improved the sentence. ‚Blood microbiota in healthy individuals persists in a latent/dormant state.‘

  1. Lines 401-402 "Numerous studies have proven the presence of microbial species in the tissues of 402 internal organs, blood, and body fluids in sick and healthy people." – how is it relevant to your study?

Answer: The tissues of internal organs, blood, and body fluids of healthy individuals are considered sterile by definition. Our investigations and studies of other authors have demonstrated that this general and dogmatic view must be subject to scientific revision.

  1. Lines 405-406: why have you selected these diseases? Do you state that arthritis, sarcoidosis, and  blood anemia have bacteria/fungal etiology?

Answer: Many publications demonstrated that arthritis, sarcoidosis, blood anemia, and many other chronic diseases have or could be associated with microbial etiology. 
